# Chronic Hepatitis B and Related Liver Diseases Are Associated with Reduced 25-Hydroxy-Vitamin D Levels: A Systematic Review and Meta-Analysis

**DOI:** 10.3390/biomedicines11010135

**Published:** 2023-01-05

**Authors:** Anindita Banerjee, Shreyasi Athalye, Naveen Khargekar, Poonam Shingade, Manisha Madkaikar

**Affiliations:** 1Transfusion Transmitted Disease Department, ICMR-National Institute of Immunohaematology, Parel, Mumbai 400012, India; 2Hematogenetics Department, ICMR-National Institute of Immunohaematology, Parel, Mumbai 400012, India; 3Department of Community Medicine, ESIC Medical College, Gulbarga 585106, India; 4Pediatric Immunology & Leukocyte Biology Department, ICMR-National Institute of Immunohaematology, Parel, Mumbai 400012, India

**Keywords:** vitamin D, chronic hepatitis B, chronic liver disease, 25-hydroxy-vitamin D, fibrosis, cirrhosis, meta-analysis

## Abstract

Hepatitis B infection is a major public health problem globally leading to chronic liver disease and death, which are influenced by various environmental and host factors including serum 25-hydroxy-vitamin D levels. There is no comprehensive systematic review reporting the association of serum 25-hydroxy-vitamin D levels and different stages of chronic hepatitis B. This study aimed to analyze the association of 25-hydroxy-vitamin D levels in chronic hepatitis B with various determinants and outcomes. A bibliographic search in PubMed, Google Scholar, and Scopus was conducted using the search terms “Vitamin D”, “cholecalciferol”, “calcitriol”, “Hepatitis B”, and “HBV”, which were published until September 2022. Meta-analysis using the “metafor” package in R was conducted with a random effect model. This analysis included 33 studies with 6360 chronic hepatitis B patients. The pooled estimates of serum 25-hydroxy-vitamin D level among CHB cases was 21.05 ng/mL and was significantly lower compared to healthy controls. (*p* < 0.005). Reduced serum 25-hydroxy-vitamin D level was significantly associated with the severity of liver fibrosis as well as HBe positivity. This analysis suggests that serum 25-hydroxy-vitamin D levels are associated with disease activity and pathobiology, although the exact nature of the cause–effect relationship cannot be discerned from this study.

## 1. Introduction

Hepatitis B infection is a public health problem affecting nearly one third of the population worldwide. In some of the infected individuals, hepatitis B infection persists as chronic infection, which later leads to complications such as hepatic fibrosis, decompensated cirrhosis, and hepatocellular carcinoma [1]. The progression of hepatitis B infection to chronic infection and liver decomposition is regulated by various host and environmental factors [1,2]. The host immune factors are controlled by nutrition, endocrine, and other determinants [1].Vitamin D is one such molecule with multiple effects on immunity, inflammation, and fibrosis. Vitamin D, a fat-soluble vitamin is crucial for a plethora of biological and physiological functions in the body. The biologically active form of vitamin D is Calcitriol (1, 25-dihydroxy vitamin D); however, the serum level of vitamin D is determined by the level of 25-hydroxy-vitamin D, which is an indicator of vitamin D sufficiency. While various classifications of vitamin D deficiency have been proposed, the widely accepted classification of vitamin D deficiency is as follows: Normal/optimum: >30 ng/mL; Insufficiency: >20 but <30 ng/mL; mild deficiency: >10 but <20 ng/mL; severe deficiency: <10 ng/mL [3]. Vitamin D3 or cholecalciferol is a hormone synthesized from the skin upon sun exposure and gets hydroxylated in the liver by 25-hydroxylase to be converted to 25-hydroxy-vitamin D followed by one more round of hydroxylation in the kidney to produce 1,25-dihydroxy-vitamin D or calcitriol, which is the physiologically active form of vitamin D. Calcitriol has a very short half-life and is not representative of the body’s reserve of vitamin D levels, which is why serum 25-hydroxy-vitamin D levels are considered as a measure of the sufficiency of vitamin D. Epidemiological evidence have highlighted the role of vitamin D in the pathogenesis of several infectious, autoimmune, and malignant diseases. Vitamin D supplementation in a few of these disease cohorts has been found to be beneficial [4]. Experimental data shows that apart from bone and muscle health and calcium metabolism, vitamin D exerts direct anti-inflammatory, anti-viral and immunomodulatory effects through various cross-talks with the cellular targets, which are activated by vitamin D-VDR signaling [5,6]. It is a well-known fact that host-viral interaction and host-immune response are responsible for various HBV outcomes. Dysregulated immune response can lead to chronic infection and late complications. Interactive crosstalk between innate and adaptive immune response largely determines the ultimate control of HBV infection. Available immunological reports found that Toll-Like Receptors (TLRs), monocytes, natural killer T-cells, cytotoxic T lymphocyte (CTL), Th1 CD4+ T cells, and dendritic cells play an essential role in the fate of HBV infection [7,8]. As vitamin D has an immunomodulatory role, it is presumable that host vitamin D levels might determine the outcome of HBV infection. Lower levels of vitamin D have been reported to be associated with chronic liver disease [9]. Moreover, one study from Israel found that patients with chronic hepatitis C with higher vitamin D levels demonstrated better virological response compared to those with lower levels [10]. Previous studies on chronic hepatitis B patients also found patients with HBV infections had significantly low levels of vitamin D and HBV DNA levels correlated inversely with the serum vitamin D levels [11]. Similarly, one meta-analysis of seven such studies found that chronic hepatitis B patients had low vitamin D levels and it correlated with the viral load [12]. However, the number of included studies was very small, with most studies from Asia. Furthermore, the study did not explore the association of vitamin D levels with disease status or other variables, which can strongly influence the association. Thus, it remains to be determined whether serum 25-hydroxy-vitamin D levels differ significantly among chronic hepatitis B patients and influence the various outcomes. Therefore, the aim of the study is to provide a detailed account of the role of 25-hydroxy-vitamin D levels in the context of chronic hepatitis B with various determinants and outcomes.

## 2. Materials and Methods

### 2.1. Protocol and Registration

The Preferred Reporting Items for Systematic Reviews and Meta-Analysis (PRISMA) 2020 guidelines were followed throughout this systematic review and meta-analysis [13]. The review protocol was registered in the International prospective register of systematic reviews, PROSPERO. (https://www.crd.york.ac.uk/prospero/display_record.php?RecordID=366783, Registration number: CRD42022366783, accessed on 15 December 2022).

### 2.2. Search Strategy

We conducted a search for studies published from January 2000 to September 2022 in PubMed, Google Scholar, and Scopus. Search terms used for identifying publications included “Vitamin D”, “Hepatitis B”, “Calcitriol”, “Cholecalciferol”, and “HBV”. The database search strategy and results are provided in Table 1.

### 2.3. Study Inclusion/Exclusion Criteria as per PICOS

#### 2.3.1. Inclusion Criteria

The following inclusion criteria were adopted to screen the articles:Individuals with hepatitis B infection (inactive carriers, chronic hepatitis);Adult participants >18 years of age;Classified the serum vitamin D concentration as either mean Vitamin D (nmol/L; ng/mL);Articles reporting hepatitis B infection with cirrhosis/fibrosis/HCC;In the English language;Full text available;Retrospective studies, cross-sectional, cohort, or randomized controlled trials.

#### 2.3.2. Exclusion Criteria

We excluded articles that were in a foreign language, or which did not contain the relevant information as per the inclusion criteria;Studies including pregnant women and children were not included;We also excluded literature reviews, editorial reviews, and systematic reviews;Editorials, brief communications, and conference proceedings were also excluded;Articles estimating the values of only 1, 25-dihydroxy vitamin D were also excluded.

### 2.4. Evaluation of the Methodological Quality of the Studies Included

The methodological quality was independently assessed by two reviewers (SA and NK) using Rayyan Software [14]. Discussion with a third reviewer (AB) was performed to resolve the disagreements and conflicts. The quality assessment was performed using the Downs and Black checklist for quality assessment [15]. The tool considers study-related characteristics of quality, external validity, biases such as study and selection bias, confounding variable consideration and power to assess the overall quality. One point (yes) or zero (no) was scored for each item, excluding the power question. The power was scored on a 6-point scale, and the post-hoc power calculations were performed using G*power software [16]. The total score determined an overall quality index, which was used to classify the studies as excellent (>25), good (18–25), fair (13–17) and poor (<13).

### 2.5. Data Extraction

SA and NK extracted the data using a standardized data format from studies that gave the number of cases according to 25-hydroxy-vitamin D levels and hepatitis B infection/outcomes. Research articles with inadequate or unclear results were excluded from quantitative analysis. Any discrepancies were resolved by through detailed discussion and consensus between the two authors (SA and NK) and an independent review by the third reviewer (AB). An electronic spreadsheet was created in which the following information was recorded: authors, year of publication, country where the study was conducted, type of publication, study design, sample size, gender, age, and 25-hydroxy-vitamin D levels (in cases and controls, in case of case-control studies).

### 2.6. Statistical Analysis

Data analysis was performed using the ‘meta’ and ‘metafor’ package in RStudio 2022.07.2 Build 576 with R for Windows version 4.2.1. The packages contain functions to estimate the common effect and random effects, generate meta-analytical plots such as forest plots, funnel plots, as well as sub-group and meta-regression analysis. A mean difference with a 95% confidence interval was used to determine the difference between the serum 25-hydroxy-vitamin D levels between chronic hepatitis B patients and controls. Cochran Q test and I^2^ statistics were used to estimate the heterogeneity between the studies. The percentage of variation across studies leading to heterogeneity rather than chance was defined as low, moderate, or high for values of 25%, 50%, and 75%, respectively. The publication bias was assessed using Funnel plots and Egger’s test.

## 3. Results

### 3.1. Search Results and Study Selection

A total of 6458 articles were identified through the search strategy, out of which 3297 were removed as duplicates and 3120 were excluded after screening titles and abstracts. Full-text review of 40 articles was performed, out of which 7 articles were excluded for not matching the eligibility criteria. A total of 33 articles were included in the meta-analysis. The details of the screening process are described in Figure 1**.**

### 3.2. Study Characteristics and Quality Assessment

The studies included in this review were published until 2022. All the studies were in English. The studies were conducted in China (*n* = 9), Iran (*n* = 4), Egypt (*n* = 3), Turkey (*n* = 3), Germany (*n* = 2), India (*n* = 2), Pakistan (*n* = 2), Poland (*n* = 2), Taiwan (*n* = 2), Israel (*n* = 1), Korea (*n* = 1), and Vietnam (*n* = 1). One study was a multi-centric study conducted in multiple countries.

Out of the studies selected for review, 18 were cross-sectional studies, 10 were case-control studies, 2 were cohort studies, and 3 were randomized controlled trials. All the studies had cases with CHB or inactive HBV carriers as the study population. One study involved patients with CTP-A cirrhosis and one study had patients with HBsAg seroclearance. Only the relevant and suitable data of CHB patients in each study were included in the meta-analysis.

As per the Downs and Black quality assessment, 8 studies were graded as good quality, 23 as fair and 2 studies were graded as poor. The agreement between the reviewers was calculated using the Cohen’s kappa statistic, and showed almost perfect agreement (99.7%) with Cohen’s k value 0.927 [17]. The study characteristics of the studies included in the review are described in Table 2.

### 3.3. Pooled Estimates

Thirty studies reported the mean 25-hydroxy-vitamin D levels. The pooled estimate for 25-hydroxy-vitamin D levels among CHB cases was 21.0568 ng/mL (95% CI: 17.5815–24.5321) (Cochran Q test *p* < 0.001, I2 = 99.4%).

### 3.4. Meta-Analysis

Meta-analysis was performed to study the difference between 25-hydroxy-vitamin D levels among cases with CHB and healthy controls. The units of 25-hydroxy-vitamin D levels were transformed from nmol/L to ng/mL to maintain uniformity in the results. As described in the forest plot (Figure 2a) The average serum 25-hydroxy-vitamin D levels were significantly lower in CHB patients than in healthy controls and the pooled mean difference was −0.59 ng/mL (−0.82, −0.35) (Cochran Q test *p* < 0.001, I2 = 91.1%). As seen in Figure 2b, the pooled mean difference of vitamin D levels between HBV carriers and healthy controls was −1.32 ng/mL (−2.84, −0.20) (Cochran Q test *p* < 0.01, I2 = 96%).

The funnel plot for 25-hydroxy-vitamin D levels in CHB versus healthy controls is asymmetrical, as shown in Figure 3a. The Egger’s test, however, suggested no evidence of potential publication bias (*p* = 0.4143).

The differences in 25-hydroxy-vitamin D levels according to HBeAg status, liver cirrhosis/fibrosis, and the effect of antiviral treatment were also analyzed. Five studies studied the differences in 25-hydroxy-vitamin D levels according to HBeAg status and found that 25-hydroxy-vitamin D levels were lower among HBeAg-positive patients as compared to HBeAg-negative patients. The pooled mean difference among the HBeAg-positive and negative groups was −0.4 ng/mL (−0.75, −0.05) (Cochran Q test *p* < 0.001, I2 = 85%). (Figure 4) The funnel plot for 25-hydroxy-vitamin D levels in HBeAg-positive versus negative cases is almost symmetrical with one outlier, as shown in Figure 3b. Five studies studied the 25-hydroxy-vitamin D levels in patients with cirrhosis and no cirrhosis, with a pooled mean difference of −0.48 ng/mL (−0.78, −0.18) (Cochran Q test *p* < 0.001, I2 = 82%). Three studies studied the 25-hydroxy-vitamin D levels in patients with fibrosis and no fibrosis with a pooled mean difference of −0.50 ng/mL (−1.86, 0.87) (Cochran Q test *p* <0.001, I2 = 85%). (Figure 5) The funnel plot for 25-hydroxy-vitamin D levels in CHB patients with or without liver disease shows studies concentrated around the top of the funnel, with only one outlier, as shown in Figure 3c. Four studies explored the difference in 25-hydroxy-vitamin D levels in treatment-naïve patients and patients on anti-viral treatment with a pooled mean difference of −0.14 ng/mL (−1.86, 0.87) (Cochran Q test *p* < 0.001, I2 = 85%) (Figure 2c).

### 3.5. Sensitivity Analysis

After excluding four studies that were lying outside the funnel plot, the standardized mean difference between CHB cases and controls was 0.3274 ng/mL [0.2104, 0.4443], which was statistically significant with a *p* value < 0.001 (Cochran Q test *p* = 0.1238, I^2^ = 38.4%)

### 3.6. Meta-Regression

We performed meta-regression analysis to see if latitude, age, male-to-female ratio among cases and control, and type of assay used for detection had any effect on the standardized mean difference of 25-hydroxy-vitamin D levels among CHB and healthy controls. As shown in Table 3, we could not find any significant association between these variables and the standardized mean difference, except that the method of detection had a significant impact in CHB cases (*p* = 0.0306). The scatterplot of latitude, methods, and gender ratio in cases versus controls was plotted as shown in Figure 6, Figure 7 and Figure 8.

## 4. Discussion

According to WHO, the current global burden of chronic hepatitis B-infected people is around 296 million people, with around 1.5 million new infections added every year [49]. A total of 33 studies were included in the present analysis, covering a total population of 6360, with 6037 for chronic hepatitis B, and 240 inactive carriers. The majority of the studies used 25-hydroxy-vitamin D to assess serum vitamin D levels in chronic hepatitis B patients. Our results indicate that chronic hepatitis B infection was associated with reduced 25-hydroxy-vitamin D levels. The role of 25-hydroxy-vitamin D in hepatitis B has been implicated by various studies, though its association with disease status has not been analyzed. This meta-analysis has explained the association of serum 25-hydroxy vitamin D levels in different stages of chronic hepatitis B patients. Our study has shown that significantly low levels of 25-hydroxy-vitamin D are associated with chronic hepatitis B patients compared to healthy controls. Furthermore, we found that this association was also found among inactive carriers of hepatitis B infection. In addition, our results also pointed out that serum levels of 25-hydroxy-vitamin D were further low in treatment-naïve patients compared to those on antiviral treatment. Thus, it would be plausible to suggest that lower 25-hydroxy-vitamin D levels can influence the viral outcome, and vice versa, pointing toward a multifaceted crosstalk among host and viral factors. As a causal factor, low serum 25hydroxy-vitamin D levels might influence the viral outcome by affecting appropriate immune response; thus, leading to chronicity. On the other hand, liver inflammation and pathology in hepatitis can compromise the 25 hydroxylation of cholecalciferol in the liver, the key step in vitamin D activation.

A study explored the effect of Vitamin D supplementation on HBV replication but did not find any significant change in HBV DNA levels before and after supplementation. Furthermore, this study studied the effect after supplementation of 2 months, and the long-term effect of supplementation on viral activity remains to be explored [48]. Another study has suggested that impaired liver function in HBV-related cirrhosis could be responsible for insufficient vitamin D hydroxylation and subsequent activation leading to reduced levels in the blood [9]. In line with this, one prospective study has found that serum 25-hydroxy-vitamin D levels negatively correlated with the severity of cirrhosis, with the lowest levels found in decompensated end-stage liver disease [50]. Moreover, as the liver is the principal organ for different transport protein synthesis, reduced production of 25-hydroxy-vitamin D binding protein in chronic hepatitis could further promote insufficient 25-hydroxy-vitamin D levels in circulation. Lastly, ethnic and geo-environmental factors such as location of residence, skin tone, seasonal variations, exposure to sunlight, and nutritional intake can influence the serum 25-hydroxy-vitamin D3 levels

Our next key observation was reduced levels of 25-hydroxy-vitamin D levels in HBe antigen-positive patients compared to HBeAg-negative ones. It highlights that 25-hydroxy-vitamin D levels affect the E antigen secretion viral proliferation in the host. In agreement with this, another study has found that 25-hydroxy-vitamin D levels inversely correlated with HBV DNA load in chronic hepatitis B patients [12].

Although a previous meta-analysis has shown the inverse correlation between 25-hydroxy-vitamin D levels and HBV DNA in chronic hepatitis B patients, no study has ever explored the detailed analysis of all studies estimating 25-hydroxy-vitamin D levels in hepatitis B patients with a different disease or treatment status. Our study has shown that 25-hydroxy-vitamin D levels varied significantly in treatment-naïve CHB patients compared to those on antiviral treatment. This again strengthens the fact that HBV infection in the liver impairs 25-hydroxy-vitamin D metabolism and its levels. Host immunological response is also governed by 25-hydroxy-vitamin D levels. A study on hepatitis B patients on interferon therapy has found that HBV DNA levels decreased more rapidly in patients with a higher level of serum 25-hydroxy-vitamin D. Thus, it signifies that 25-hydroxy-vitamin D influences the cellular immune response to a great extent [51].

Furthermore, results from subgroup analysis among CHB patients with fibrosis and cirrhosis indicated that 25-hydroxy-vitamin D levels positively correlated with fibrosis severity by the common effects model [*p* < 0.01]. A study administering ergo/cholecalciferol in alcoholic liver disease patients has demonstrated that the biological availability of 25-hydroxy-vitamin D positively correlated with patients with mild to moderate liver fibrosis compared with patients with severe liver disease (Child Pugh-C) [52]. The negative correlation between the severity of fibrosis/cirrhosis and 25-hydroxy-vitamin D levels could also be due to the fact that vitamin D, through the activation (VDR) and Calcium-sensing receptor (CaSR), imparts portal hypotensive effect as evident from a study in a rat model [53].

Meta-regression was performed to analyze how common variables such as age, sex, latitude, and the method of 25-hydroxy-vitamin D estimation influenced or showed any association with the serum 25-hydroxy-vitamin D level (intervention effect) in the meta-analysis. The analysis showed no significant association between these variables with the estimation of serum Vitamin D levels in cases vs. healthy control, suggesting these factors were adjusted and nullified.

However, the study has some limitations which should be considered while interpreting the results. The heterogeneity among studies was quite high due to different study types, sample size, and different disease states related to chronic HBV. There was also significant publication bias as evident from our findings. However, sensitivity analysis was performed to check the robustness of the results and models and it was found that after excluding four studies the mean difference remained statistically significant. Lastly, although we tried to incorporate all eligible published studies, there are still chances of missing a few studies from the grey literature.

## 5. Conclusions

The meta-analysis has covered the most updated and pooled estimate of serum 25-hydroxy-vitamin D levels in altogether and different disease states of chronic hepatitis B patients. Presumably, this is the first systematic review and meta-analysis which has identified major differences in serum 25-hydroxy-vitamin D levels correlating with disease activity such as HBeAg status and severity in terms of fibrosis and cirrhosis. Notwithstanding the wide heterogeneity among the included studies, our analysis strongly suggests that serum 25-hydroxy-vitamin D levels are associated with disease activity and pathobiology, although the exact nature of the cause-effect relationship cannot be discerned from this study. Future research is necessary to conduct in this area to validate the therapeutic and preventive role of vitamin D against chronic hepatitis B and related liver diseases.

Thus, this detailed meta-analysis provides corroborative evidence about the role of vitamin D in hepatitis B-related diseases and suggests that monitoring 25-hydroxy-vitamin D status and supplementation of vitamin D in chronic hepatitis B patients to prevent long-term complications should be addressed in future well-designed study cohorts.

## Figures and Tables

**Figure 1 biomedicines-11-00135-f001:**
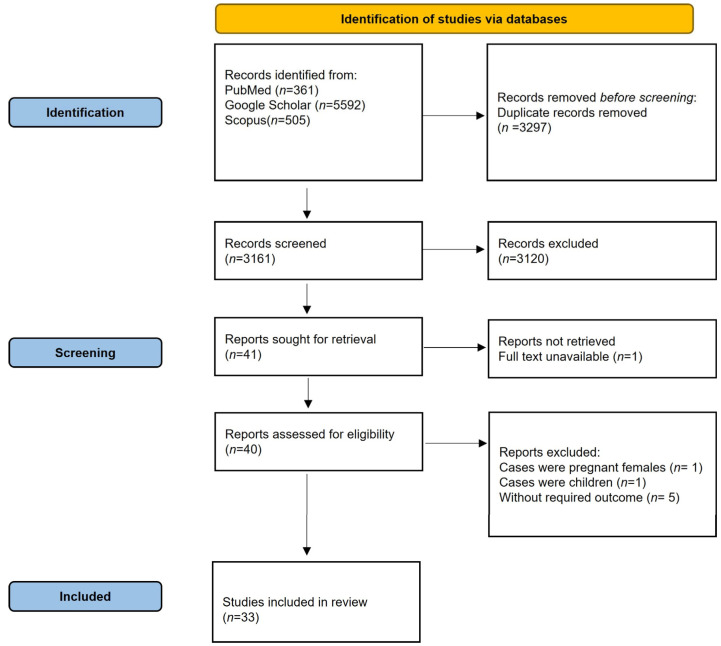
PRISMA 2020 flow diagram of the study selection process.

**Figure 2 biomedicines-11-00135-f002:**
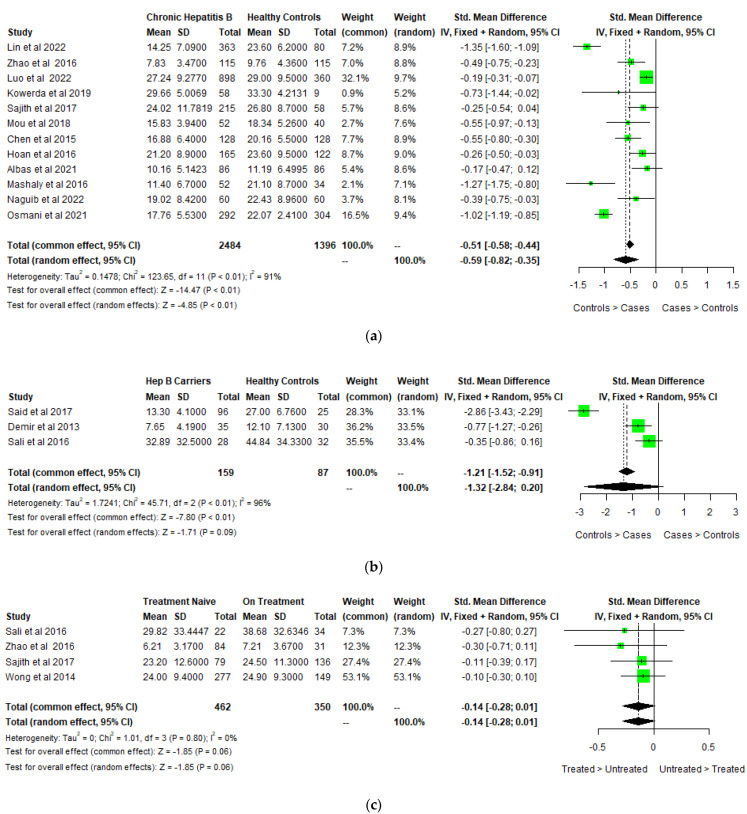
Forest plot of vitamin D levels between (**a**) CHB cases and healthy controls [9,21,22,23,24,25,26,27,28,29,30,31], (**b**) inactive HBV carriers and healthy controls [18,19,20], (**c**) treatment-naïve and on-treatment Hepatitis B patients [20,22,25,41].

**Figure 3 biomedicines-11-00135-f003:**
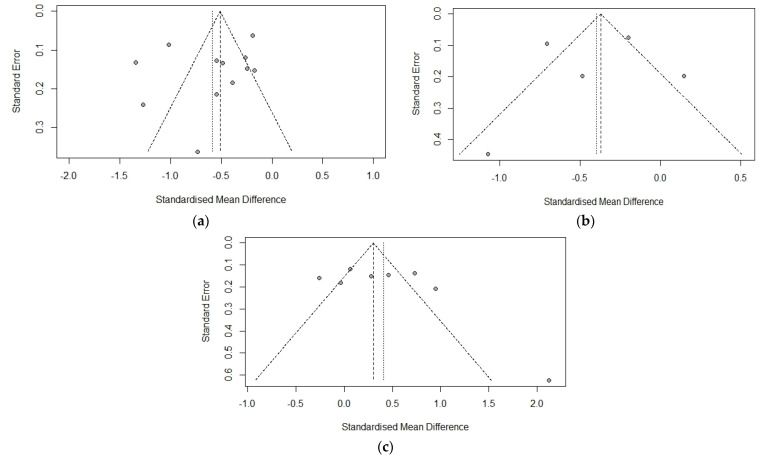
Funnel plot for publication bias analysis for vitamin D levels between (**a**) CHB patients and healthy controls (**b**) HBeAg-positive and negative CHB patients (**c**) CHB patients with or without liver disease.

**Figure 4 biomedicines-11-00135-f004:**
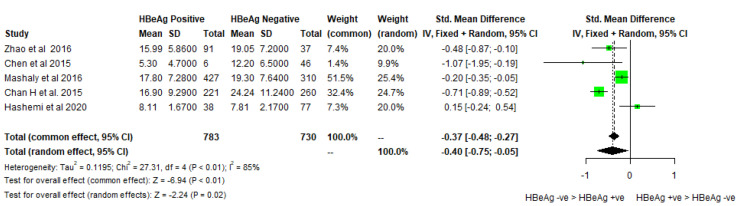
Forest plot of vitamin D levels between HBeAg-positive and HBeAg-negative Chronic Hepatitis B cases [22,27,29,36,43].

**Figure 5 biomedicines-11-00135-f005:**
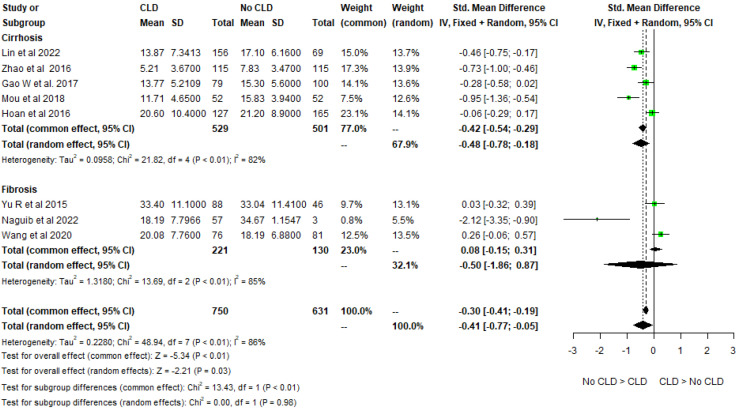
Forest plots of vitamin D levels in CHB patients with absence or presence of cirrhosis [9,21,22,26,37]/fibrosis [30,35,48].

**Figure 6 biomedicines-11-00135-f006:**
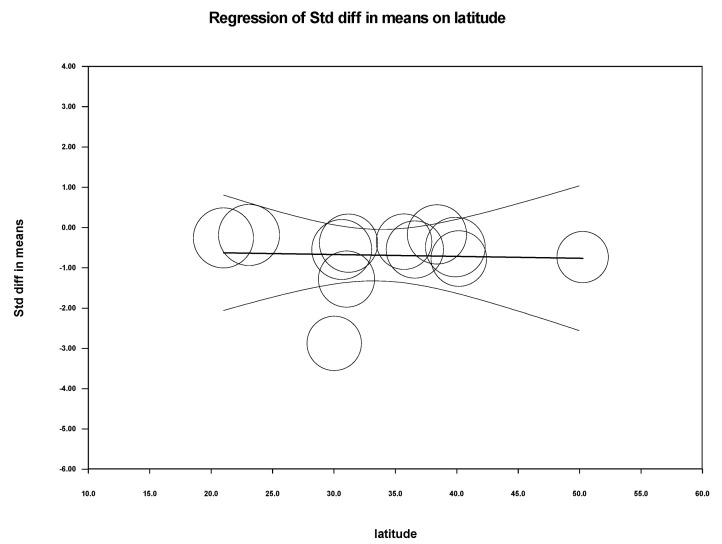
Cases vs. Controls: Scatterplot for latitude.

**Figure 7 biomedicines-11-00135-f007:**
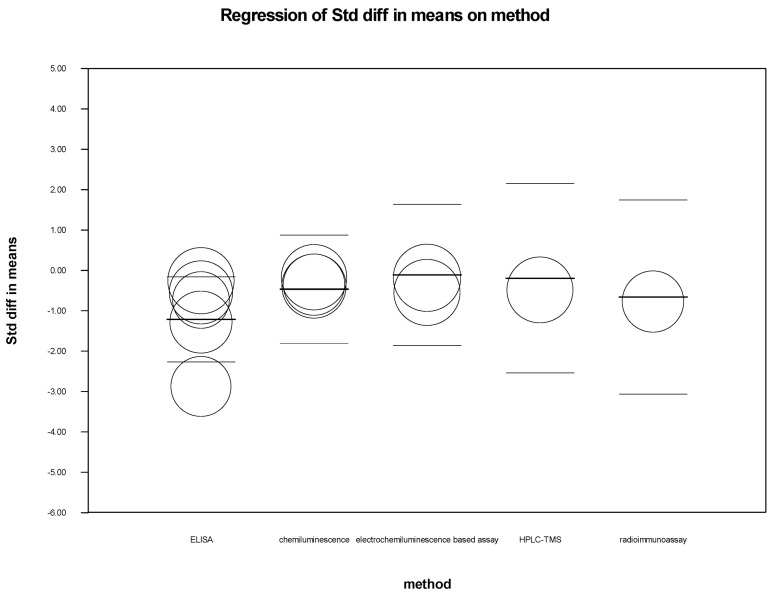
Cases vs. Controls: Scatterplot for method.

**Figure 8 biomedicines-11-00135-f008:**
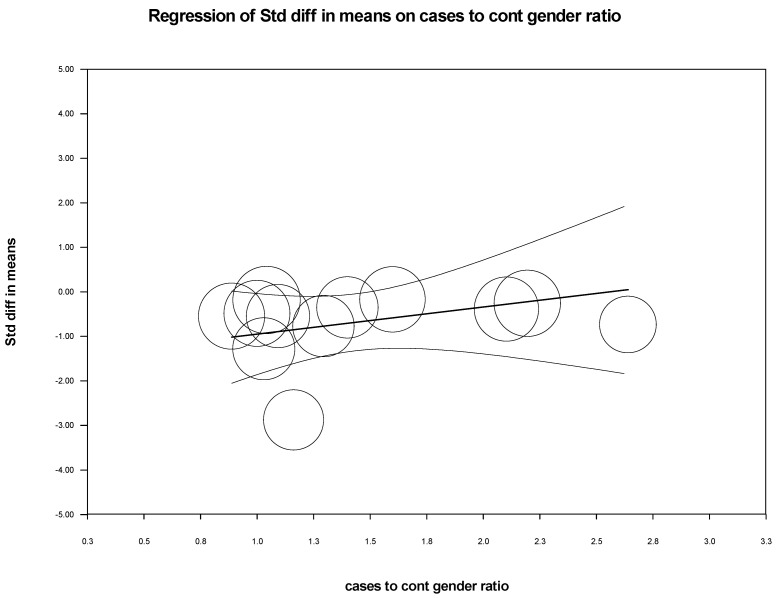
Cases vs. Controls: Scatterplot for Gender Ratio in Cases to Controls.

**Table 1 biomedicines-11-00135-t001:** Database search strategy and results.

Sr. No.	Key Words	Databases		
		PubMed	Google Scholar	Scopus
1	“Calcitriol” and “Hepatitis B”	44	998	132
2	“Cholecalciferol” and “Hepatitis B”	6	999	8
3	“Vitamin D” and “Hepatitis B”	192	998	200
4	“Calcitriol” and “HBV”	26	996	32
5	“Cholecalciferol” and “HBV”	2	606	3
6	“Vitamin D” and “HBV”	91	995	130

The publications were uploaded in Rayyan software (https://rayyan.ai/, accessed on 15 December 2022) to remove the duplicates and screen the articles according to the inclusion criteria [14].

**Table 2 biomedicines-11-00135-t002:** Main characteristics of included studies.

Author & Year	Country	Study Design	Number of Cases	CHB State	Number of Healthy Controls	Vitamin D Estimation Method	Vitamin D in Patients (Mean ± SD) (ng/mL)	Vitamin D in Controls (Mean ± SD) (ng/mL)	Quality Score
Said et al. 2017 [18]	Egypt	retrospective case control	96	Inactive carriers	25	ELISA	13.3 ± 4.1	27 ± 6.76	17
Demir et al. 2013 [19]	Turkey	cross-sectional	35	Inactive carriers	30	radioimmunoassay	7.65 ± 4.19	12.1 ± 7.13	18
Sali et al. 2016 [20]	Iran	retrospective case control	28	Inactive carriers	32	chemiluminescence	32.89 ± 32.5	44.84 ± 34.33	13
Lin et al. 2022 [21]	China	retrospective case control	363	CHB	80	electrochemiluminescence based assay	14.25 ± 7.09	23.6 ± 6.2	22
Zhao et al. 2016 [22]	China	retrospective case control	115	CHB	115	HPLC-TMS	7.83 ± 3.47	9.76 ± 4.36	21
Luo et al. 2022 [23]	China	cross-sectional	898	CHB & MAFLD with CHB	360	electrochemiluminescence based assay	27.2371 ± 9.277	29 ± 9.5	20
Kowerda et al. 2019 [24]	Poland	retrospective case control	58	CHB	9	ELISA	29.6646 ± 5.0069	33.2984 ± 4.2131	15
Sajith et al. 2017 [25]	India	retrospective case control	215	CHB	58	chemiluminescence	24.0223 ± 11.7819	26.8 ± 8.7	16
Mou et al. 2018 [26]	China	retrospective case control	52	CHB	40	ELISA	15.83 ± 3.94	18.34 ± 5.26	16
Chen et al. 2015 [27]	China	cohort	128	CHB	128	electrochemiluminescence based assay	16.88 ± 6.4	20.16 ± 5.5	22
Hoan et al. 2016 [9]	Vietnam	cross-sectional	165	CHB	122	ELISA	21.2 ± 8.9	23.6 ± 9.5	17
Albas et al. 2021 [28]	Turkey	cross-sectional	86	CHB	86	chemiluminescence	10.16 ± 5.1423	11.1895 ± 6.4995	16
Mashaly et al. 2016 [29]	Egypt	cross-sectional	52	CHB	34	ELISA	11.4 ± 6.7	21.1 ± 8.7	19
Naguib et al. 2022 [30]	Egypt	retrospective case control	60	CHB	60	chemiluminescence	19.02 ± 8.42	22.43 ± 8.96	14
Osmani et al. 2021b [31]	Iran	retrospective case control	292	CHB	304	electrochemiluminescence based assay	17.76 ± 5.53	22.07 ± 2.41	20
Thakur et al. 2021 [32]	India	cross-sectional	30	Hep B Cirrhosis	30	electrochemiluminescence based assay	25.4 ± 11	30.4 ± 8.6	16
Mahamid et al. 2013 [33]	Israel	cross-sectional	53	HBsAg seroclearance	-	NA	28.0283 ± 8.0753	-	16
Yu R et al. 2018 [34]	China	RCT	560	CHB	-	electrochemiluminescence based assay	29.64 ± 11.29	-	15
Yu R et al. 2015 [35]	China	cross-sectional	242	CHB	-	electrochemiluminescence based assay	33.9 ± 10.67	-	16
Chan H et al. 2015 [36]		RCT	737	CHB	-	chemiluminescence	18.4 ± 7.46	-	16
Gao W et al. 2017 [37]	China	cross-sectional	100	CHB	-	NA	15.3 ± 5.6	-	16
Farnik et al. 2013 [11]	Germany	retrospective case control	203	CHB	-	radioimmunoassay	14.4 ± 7.9	-	16
Osmani et al. 2021a [38]	Iran	cross-sectional	292	CHB	-	electrochemiluminescence based assay	18.4 ± 3.5	-	16
Ko et al. 2016 [39]	Korea	cross-sectional	207	CHB	-	isotope-dilution liquid chromatography-tandem mass spectrometry.	13.4717 ± 7.1565	-	16
Ko et al. 2020 [40]	Taiwan	cross-sectional	60	CHB	-	chemiluminescence	20.9 ± 5.6	-	16
Wong et al. 2014 [41]	China	cohort	426	CHB	-	electrochemiluminescence based assay	24.3 ± 9.4	-	16
Berkan-Kawinska et al. 2015 [42]	Poland	cross-sectional	35	CHB	-	chemiluminescence	17.6 ± -	-	15
Hashemi et al. 2020 [43]	Iran	cross-sectional	281	CHB	-	ELISA	23.69 ± 11.26	-	15
Karim et al. 2021 [44]	Pakistan	cross-sectional	108	CHB	-	chemiluminescence	25.23 ± -	-	10
Kumar et al. 2021 [45]	Pakistan	cross-sectional	93	CHB	-	NA	24.31 ± -	-	10
Motor et al. 2014 [46]	Turkey	cross-sectional	81	Inactive carriers	-	chemiluminescence	52.764 ± 20.03	-	13
Schiefke et al. 2005 [47]	Germany	cross-sectional	13	CHB	-	biochemistry assay	31.2354 ± 13.3896	-	13
Wang et al. 2020 [48]	Taiwan	RCT	196	CHB	-	chemiluminescence	19.8 ± 7.4	-	21

RCT: Randomised Controlled Trial, CHB: Chronic Hepatitis B, MAFLD: Metabolic-associated fatty liver disease, HPLC-TMS: High Performance Liquid Chromatography Tandem Mass Spectrometry method, NA: Not Available.

**Table 3 biomedicines-11-00135-t003:** Random effect meta-regression analysis of 25-hydroxy-vitamin D levels.

Covariate	Coefficient	Standard Error	95% Lower	95% Upper	Z-Value	2-Sided *p*-Value	Set
**Cases vs. Controls**							
Intercept	−1.9311	1.0683	−4.025	0.1628	−1.81	0.0707	
Latitude	−0.0047	0.0273	−0.0581	0.0488	−0.17	0.8635	
Chemiluminescence Method	0.7477	0.4402	−0.1152	1.6105	1.7	0.0894	Q = 6.03, df = 4, *p* = 0.1971
Electrochemiluminescence Method	1.1013	0.5598	0.004	2.1985	1.97	0.0492	
HPLC-TMS Method	1.0181	0.7092	−0.3718	2.4081	1.44	0.1511	
Radioimmunoassay Method	0.5537	0.7118	−0.8414	1.9488	0.78	0.4366	
Cases to Controls Gender Ratio	0.6102	0.3962	−0.1663	1.3867	1.54	0.1235	
**All Cases**							
Intercept	53.1721	29.3444	−4.3418	110.686	1.81	0.07	
Chemiluminescence Method	−3.2579	10.6526	−24.1366	17.6208	−0.31	0.7597	Q = 13.91, df = 6, *p* = 0.0306
Electrochemiluminescence Method	−9.2847	11.8277	−32.4666	13.8971	−0.78	0.4325	
ELISA Method	−13.5263	10.6162	−34.3337	7.2811	−1.27	0.2026	
HPLC-TMS Method	−19.5501	12.7232	−44.4871	5.3868	−1.54	0.1244	
Isotope-dilution Liquid Chromatography-tandem Mass Spectrometry Method	−15.438	12.1771	−39.3047	8.4288	−1.27	0.2049	
Radioimmunoassay Method	−27.7252	11.4111	−50.0906	−5.3598	−2.43	0.0151	
Latitude	0.2235	0.2937	−0.3521	0.7991	0.76	0.4466	
Age	−0.6979	0.4296	−1.5398	0.1441	−1.62	0.1043	
Male female ratio	0.3166	2.1212	−3.8409	4.474	0.15	0.8814	
**All Controls**							
Intercept	−71.6418	82.2164	−232.783	89.4993	−0.87	0.3835	
Latitude	0.9468	0.8782	−0.7744	2.668	1.08	0.281	
Electrochemiluminescence Method	−9.1972	16.0905	−40.734	22.3395	−0.57	0.5676	Q = 3.09, df = 4, *p* = 0.5421
ELISA Method	−2.4554	8.7522	−19.6094	14.6985	−0.28	0.7791	
HPLC-TMS Method	−60.6786	43.2949	−145.535	24.1777	−1.4	0.1611	
Radioimmunoassay Method	−8.1006	15.618	−38.7113	22.5101	−0.52	0.604	
Male female ratio	14.047	13.5259	−12.4632	40.5572	1.04	0.299	
Age	1.0935	1.0572	−0.9785	3.1655	1.03	0.3009	

## Data Availability

All relevant data were included in the paper.

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
