# Peer review of "Chronic Hepatitis B and Related Liver Diseases Are Associated with Reduced 25-Hydroxy-Vitamin D Levels: A Systematic Review and Meta-Analysis"

_biomedicines, 2023, doi:10.3390/biomedicines11010135_

Round 1

Reviewer 1 Report

This original review arouses interest for readers and provides an important clue to discuss the association between the vitamin D level and chronic HBV infection. However, there are some problems that should be addressed or altered.

1) In general, it is well known that vitamin D levels decline as the disease stage/status or liver fibrosis progresses, not only in hepatitis B, but in other liver diseases (such as chronic HCV infection, AIH, and PBC) as well. As described by authors, liver dysfunction impairs vitamin D metabolism. Therefore, it is conceivable that serum/plasma vitamin D levels can be decreased in a stepwise manner from healthy control to inactive HBV infection, treatment-naïve active status, and on-treatment state; or from less-advanced stage to compensated and decompensated liver cirrhosis. Therefore, it is reasonable to assume that vitamin D levels merely reflect disease stage, although vitamin D can influence host immune. Are there any studies that vitamin D supplementation can improve long-term virological or clinical outcomes in chronic HBV infection? If no report, please state this issue clearly in the text. In addition, authors should specify the presence or absence of compensated/decompensated liver and NA (DAA) treatment in the revised version.

2) The descriptions in the text and the figures do not match. Authors should describe the figure legends in more detail and carefully.

3) Table 1: Why did PubMed include the term of “liver fibrosis”?

4) There are too many typographic and grammatical errors. For instance, in the introduction section, line 49, d >>> D; line 53, role “of” vitamin D; line 63, “and” dendric cells; line 66, has >>> have; line 79, various determinant“s”…. Authors should brush up their manuscript more carefully.

Author Response

  • In general, it is well known that vitamin D levels decline as the disease stage/status or liver fibrosis progresses, not only in hepatitis B, but in other liver diseases (such as chronic HCV infection, AIH, and PBC) as well. As described by authors, liver dysfunction impairs vitamin D metabolism. Therefore, it is conceivable that serum/plasma vitamin D levels can be decreased in a stepwise manner from healthy control to inactive HBV infection, treatment-naïve active status, and on-treatment state; or from less-advanced stage to compensated and decompensated liver cirrhosis. Therefore, it is reasonable to assume that vitamin D levels merely reflect disease stage, although vitamin D can influence host immune. Are there any studies that vitamin D supplementation can improve long-term virological or clinical outcomes in chronic HBV infection? If no report, please state this issue clearly in the text. In addition, authors should specify the presence or absence of compensated/decompensated liver and NA (DAA) treatment in the revised version.

Reply: We thank the reviewer for his/her careful observations and valuable suggestions. As per our knowledge, there is one clinical trial data (Wang et al, 2020) which assessed the effect of Vitamin D supplementation on Hepatitis B viral replication. This randomized controlled trial showed that although the Vitamin D levels increased after 2 months of supplementation among HBV infected individuals, it did not affect the levels of HBV DNA or HBsAg levels in circulation. The study concluded with no causal relationship between Vitamin D and HBV replication. However, the study did not enroll patients with chronic HBV patients with liver cirrhosis and also did not assess the long-term effect or clinical outcome of supplementation in them.

 The data obtained from the selected articles has been categorized on the basis of presence/absence of cirrhosis and the comparison of the same has been shown in figure 5 in the revised version of the manuscript. Only four articles discussed the Vitamin D levels among treatment naïve and on treatment patients, which has been shown in figure 2(c) in the revised version of the manuscript.

  • The descriptions in the text and the figures do not match. Authors should describe the figure legends in more detail and carefully.

Reply: We thank the reviewer for his/her careful observations. The necessary changes have been made and highlighted in the revised manuscript.

  • Table 1: Why did PubMed include the term of “liver fibrosis”?

Reply: We have revised the search strategy and key words search is same in Pubmed, Google scholar and Scopus. The updated search strategy has been incorporated in the revised manuscript and highlighted.

  • There are too many typographic and grammatical errors. For instance, in the introduction section, line 49, d >>> D; line 53, role “of” vitamin D; line 63, “and” dendric cells; line 66, has >>> have; line 79, various determinant“s”…. Authors should brush up their manuscript more carefully.

Reply: All the typographic and grammatical errors have been rectified and incorporated in the revised version of the manuscript.

Reviewer 2 Report

Could you specify the reasons behind the timeframe chosen?

The search strategy appears to be very poor. Why did you only select HBV instead of also adding hepatitis or other synonyms? the same also for vitamin D. Please clarify. Correctly selecting keywords is an essential component for conducting a high-quality systematic review. As you can notice, using this search strategy, only 12 articles were retrieved in Scopus and a bit more than 100 in Pubmed. Very limited number. Please clarify.

Please, define inclusion/exclusion criteria according to PICOS as suggested by Cochrane.

The PRISMA guidelines have been updated in 2020. Please use those guidelines and the relative flow diagram. 

Author Response

  • Could you specify the reasons behind the timeframe chosen?

Reply: The time frame of 2000 to 2022 was chosen to include the recent articles in last two decades. But now we have searched all the articles till 2022 and have included them in the review.

  • The search strategy appears to be very poor. Why did you only select HBV instead of also adding hepatitis or other synonyms? the same also for vitamin D. Please clarify. Correctly selecting keywords is an essential component for conducting a high-quality systematic review. As you can notice, using this search strategy, only 12 articles were retrieved in Scopus and a bit more than 100 in Pubmed. Very limited number. Please clarify.

Reply: We appreciate the comments by the reviewers. We have done the required changes in the search strategy and included articles with other synonym key words. However, most of them were duplicates, and our team used Rayyan software to sort out the articles.

Sr.No

Key Words

Databases

Pubmed

Google Scholar

Scopus

1

“Calcitriol” AND “Hepatitis B”

44

998

132

2

“Cholecalciferol” AND “Hepatitis B”

6

999

8

3

“Vitamin D” AND “Hepatitis B”

192

998

200

4

“Calcitriol” AND “HBV”

26

996

32

5

“Cholecalciferol” AND “HBV”

2

606

3

6

“Vitamin D” AND “HBV”

91

995

130

A total of 6458 articles were identified through search strategy, out of which 3297were removed as duplicates and 3120 were excluded after screening titles and abstracts. Full text review of 40 articles was done out of which 7 articles were excluded for not matching the eligibility criteria. Total 33 articles were included in the meta-analysis. The updated information has been incorporated in the revised manuscript and highlighted.

  • Please, define inclusion/exclusion criteria according to PICOS as suggested by Cochrane.

Reply: The Changes in the inclusion/exclusion criteria has been done according to PICOS as suggested by Cochrane and highlighted in the revised manuscript.

Study Inclusion Criteria:The following inclusion criteria was adopted to screen the articles, which are:

  1. Individuals with Hepatitis B infection (inactive carriers, chronic hepatitis)
  2. Adult participants >18 years of age
  3. classified the serum Vitamin D concentration as either mean Vitamin D (nmol/l; ng/ml),
  4. Articles reporting hepatitis b infection with cirrhosis/fibrosis/HCC
  5. In English language,
  6. Full text available,
  7. Retrospective studies, cross-sectional, cohort, or randomized controlled trials.

Exclusion Criteria:

  1. We excluded articles that were in foreign language, or which did not contain the relevant information as per the inclusion criteria.
  2. Studies including pregnant women and children were not included.
  3. We excluded also literature reviews, editorial reviews, or systematic reviews.
  4. Editorials, brief communications and conference proceedings were also excluded.
  5. Articles estimating the values of only 1,25hydroxy Vitamin D were also excluded.

  • The PRISMA guidelines have been updated in 2020. Please use those guidelines and the relative flow diagram. 

Reply: The revised Flow diagram has been reported as per PRISMA 2020 guidelines (Figure 1 of the revised manuscript).

Reviewer 3 Report

Dear Authors, I have read with a great pleasure your article describing the meta-analysis of vitamin D and hepatitis B. In my opinion, the meta-analysis has been properly planned, performed, analysed and discussed. Your conclusions are interesting and give an input for new experiments in this issue. I recommend to accept the article in present form.

Author Response

Dear Authors, I have read with a great pleasure your article describing the meta-analysis of vitamin D and hepatitis B. In my opinion, the meta-analysis has been properly planned, performed, analysed and discussed. Your conclusions are interesting and give an input for new experiments in this issue. I recommend to accept the article in present form.

Reply: We thank the reviewer for his/her valuable comments.

Round 2

Reviewer 2 Report

I am satisfied with the changes provided by the authors